# On the Equivalence of Causal Models:
# A Category-Theoretic Approach

**Jun Otsuka**                                                                 JOTSUKA@BUN.KYOTO-U.AC.JP
*Philosophy Dept., Kyoto University, Kyoto, Japan.*

**Hayato Saigo**                                                            HARMONIAHAYATO@GMAIL.COM
*Dept. of Bioscience, Nagahama Institute of Bio-Science and Technology, Shiga, Japan.*

**Editors:** Bernhard Schölkopf, Caroline Uhler and Kun Zhang

## Abstract

We develop a category-theoretic criterion for determining the equivalence of causal models having different but homomorphic directed acyclic graphs over discrete variables. Following Jacobs et al. (2019), we define a causal model as a probabilistic interpretation of a causal string diagram, i.e., a functor from the "syntactic" category $\mathsf{Syn}_G$ of graph $G$ to the category $\mathsf{Stoch}$ of finite sets and stochastic matrices. The equivalence of causal models is then defined in terms of a natural transformation or isomorphism between two such functors, which we call a $\Phi$-abstraction and $\Phi$-equivalence, respectively. It is shown that when one model is a $\Phi$-abstraction of another, the intervention calculus of the former can be consistently translated into that of the latter. We also identify the condition under which a model accommodates a $\Phi$-abstraction, when transformations are deterministic.

**Keywords:** Causal Models, Abstraction, Category Theory, String Diagrams

## 1. Introduction

Causal models offer a general framework for studying causal structures over variables. The framework, however, lacks a formal criterion as to when two causal models having different variables or graphs are nevertheless considered to be the "same." This raises an issue when one wants to extrapolate a causal model from one system to another and claim that two numerically distinct systems (say, the brain networks of macaques and humans) share the same causal structure to some extent. Alternatively, one and the same physical system may be modeled in terms of different set of variables at different granularities or levels (Chalupka et al., 2014, 2016, see also Fig. 1). Recent studies attempt to answer this question in terms of variable transformations (Rubenstein et al., 2017; Beckers and Halpern, 2019; Beckers et al., 2020), but the proposed criteria are relative to a particular sequence of interventions (as opposed to the general feature of the model) and do not reflect the topological features of the graph, which are central to causal modeling.

In this paper we propose a novel criterion and systematic method for determining the equivalence of two causal models, drawing on the category-theoretic formulation of causal models developed by Jacobs et al. (2019). In this framework, a causal model is identified with a functor, which is a probabilistic interpretation of a string diagram constructed from the directed acyclic graph (DAG) of the model. We then define the equivalence or abstraction of causal models, called a $\Phi$-abstraction, in terms of a *natural transformation* between such functors based on homomorphic DAGs. In contrast to previous approaches, a $\Phi$-abstraction is a relation between two causal models, defined without regard to a particular sequence of interventions. Interventions at different

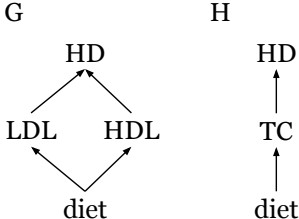

Figure 1: An example of two causal models $G$ and $H$ that describe the (supposedly) same phenomenon, the effects of diet on heart disease (HD) (adapted from Rubenstein et al., 2017). In graph $G$, diet affects heart disease through two types of blood cholesterol, low-density lipoprotein (LDL) and high-density lipoprotein (HDL). Graph $H$ combines these two variables into the total cholesterol (TC).

levels are then derived consistently from models related by the $\Phi$-abstraction, by way of a monoid homomorphism. We will also provide a necessary and sufficient condition for a given model to have a corresponding high-level abstraction, which has been absent in previous studies. While our explication focuses on the micro-macro descriptions of the same phenomena, our formulation of $\Phi$-abstraction is general and applicable to the equivalence of causal models of physically different systems.

The paper unfolds as follows. Section 2 briefly explains how to represent discrete causal models using string diagrams and functors. Section 3 then defines the equivalence and abstraction relationship between distinct causal models with homomorphic DAGs in terms of a natural equivalence and transformation. Section 4 deals with interventions, and shows that the intervention calculus of a low-level model, expressed as a monoid action, is related to that of a high-level model via a monoid homomorphism. This means that an intervention on the former is consistently translated to that on the latter. Section 5 compares our proposal with the existing approaches by Rubenstein et al. (2017) and Beckers and Halpern (2019), and shows that ours incorporates some of the previous results. A problem with the previous criteria is that they do not tell us when a given model accommodates abstraction. Section 6 explores this problem and determines a necessary and sufficient condition for a given model to have a non-trivial $\Phi$-abstraction when the transformations are deterministic. We conclude in Section 7 with a discussion of the advantage of adopting a category-theoretic approach in addressing this kind of problem.

## 2. Categorical Representation of Causal Models

In this section we briefly sketch the category-theoretic formulation of causal models. Due to lack of space, we omit technical details that have no bearing on the following discussion. We refer the reader to Jacobs et al. (2019) for the details and to Awodey (2010) or Leinster (2014) for general introductions to category theory. In their approach, a causal graph is reformulated as a string diagram category representing the "syntactical" structure of the graph, while specific causal models are regarded as "semantic" assignments of values and stochastic matrices to each component of the string diagram, i.e., functors from the string diagram category to the category of stochastic matrices Stoch.

Let $G = (V_G, E_G)$ be a DAG with discrete (categorical) variables $V_G$ and edges $E_G$. From this one can construct a string diagram category $\mathsf{Syn}_G$ whose objects are generated by the vertices of $G$, and whose morphisms are generated by the following "box" signature:

$$\Sigma_G = \left\{ \begin{array}{c} {\scriptstyle |Y} \\ \boxed{y} \\ {\scriptstyle X_1|\cdots|X_k} \end{array} \middle| X_1, \ldots, X_k \in \mathrm{PA}(Y), Y \in V_G \right\}$$

where $\mathrm{PA}(Y)$ is the set of parents of $Y$. Intuitively, each box represents a causal "mechanism" that determines its effect from the input wires/variables. A causal string diagram is constructed by combining these mechanisms as in Fig. 2, which illustrates a string diagram rendering of the graph $G$ in Fig 1. Note that in string diagrams, variables (objects) are denoted by strings and arrows by boxes, opposite to the notation in conventional causal graphs. It is assumed that the direction of causal influence flows from bottom to top.

Another category we need is the category $\mathsf{Stoch}$, whose objects are finite sets and whose morphisms $f : X \to Y$ are $|X| \times |Y|$ dimensional stochastic matrices, i.e., matrices of positive numbers whose columns each sum up to 1. Intuitively, each object (finite set) in $\mathsf{Stoch}$ represents a set of values of a particular variable, while a morphism (stochastic matrix) represents conditional probabilities for the values of an effect given its causes. A parentless (exogenous) variable $Y$ has a morphism from the object 1; this morphism is a $1 \times |Y|$ stochastic matrix or vector, and thus gives $P(Y)$, the marginal distribution of $Y$.

With this setup, a particular causal model is given by a systematic assignment that maps objects (strings) in $\mathsf{Syn}_G$ to those (finite sets of values) in $\mathsf{Stoch}$, and morphisms (boxes) in $\mathsf{Syn}_G$ to those (stochastic matrices) in $\mathsf{Stoch}$. This defines a causal model as a functor $F_G : \mathsf{Syn}_G \to \mathsf{Stoch}$. Taking Fig. 2 as an example, a functor $F_G$ assigns to each string/object a set of possible values, say, $F_G :: \mathrm{diet} \mapsto \{poor, good\}, \mathrm{LDL} \mapsto \{high, low\}$ etc. To the box below LDL, it assigns conditional probabilities $P(\mathrm{LDL}|\mathrm{diet})$ for each value of LDL and diet; these conditional probabilities can be represented by a $2 \times 2$ stochastic matrix. In this way, a functor $F_G$ represents a specific Bayesian network with graph $G$ as in Fig. 1, and conversely, any finite Bayesian network on the DAG $G$ can be represented by a functor of type $\mathsf{Syn}_G \to \mathsf{Stoch}$ (Jacobs et al., 2019, Proposition 3.1), which justifies our identification of a causal model with a functor $F_G$.

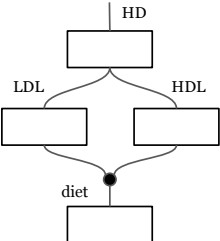

Figure 2: The string diagram rendering of graph $G$ in Fig. 1. The black dot is a "copier" that copies the values of the "diet" string (not discussed in the text).

## 3. Equivalence of Causal Models

We now consider the equivalence and abstraction of causal models in the above framework. In contrast to previous approaches (Rubenstein et al., 2017; Beckers and Halpern, 2019) that focus only on probabilistic consistency before and after transformations, we require that a transformation between causal models preserve the graphical structure, i.e., that the models' graphs are homomorphic. Let $G, H$ be DAGs, and $\phi : G \to H$ be a graph homomorphism, i.e., a function $\phi : V_G \to V_H$ such that $X \to Y \in E_G$ implies $\phi(X) \to \phi(Y) \in E_H$. Since multiple variables in $V_G$ may be mapped to a single variable in $V_H$ by $\phi$, we call the causal model based on $G$ the original/micro/low-level model, and the causal model based on $H$ the target/macro/high-level model. Let $\mathsf{Syn}_G, \mathsf{Syn}_H$ be string diagram categories, each constructed from $G$ and $H$, and $F_G, F_H$ be causal models, that is, functors from $\mathsf{Syn}_G, \mathsf{Syn}_H$ to $\mathsf{Stoch}$, respectively. Then the graph homomorphism $\phi$ naturally induces a functor $\Phi : \mathsf{Syn}_G \to \mathsf{Syn}_H$, which sends an object (string) $Y$ in $\mathsf{Syn}_G$ to object $\phi(Y)$ in $\mathsf{Syn}_H$, and boxes:

$$
\begin{array}{ccc}
\begin{array}{c} \big| Y \\ \boxed{y} \\ X_1 \big| \quad \big| X_k \end{array}
&
\mapsto
&
\begin{array}{c} \big| \phi(Y) \\ \boxed{\phi(y)} \\ \phi(X_1) \ \phi(X_k) \, Z_1 \quad \big| Z_l \end{array}
\end{array}
$$

where $Z_1 \ldots Z_l \in \mathrm{PA}(\phi(Y)) \setminus \phi(\mathrm{PA}(Y))$.

The graph homomorphism $\phi : G \to H$, along with the induced syntactical functor $\Phi : \mathsf{Syn}_G \to \mathsf{Syn}_H$, assures only the consistency of the graphical properties (i.e., cause-effect relationships) of $G$ and $H$. A transformation of causal *models* further requires the consistency of their probability assignment to variables, which in the present categorical framework amounts to the consistency of functors to $\mathsf{Stoch}$. This consistency condition is given by the following notion of a $\Phi$-abstraction.

**Definition 1 ($\Phi$-abstraction)** *Let $\phi : G \to H$ be a graph homomorphism; $\Phi : \mathsf{Syn}_G \to \mathsf{Syn}_H$ the induced functor; and $F_G, F_H$ functors (causal models) to $\mathsf{Stoch}$ from $\mathsf{Syn}_G$ and $\mathsf{Syn}_H$, respectively. We say that $F_H$ is a $\Phi$-abstraction of $F_G$ if there is a natural transformation $\alpha : F_G \Rightarrow F_H \Phi$.*

In category theory, a natural transformation is a set of morphisms that relate two functors in a consistent fashion. In the present case, the natural transformation $\alpha$ is a set of morphisms in $\mathsf{Stoch}$, i.e., stochastic matrices whose entries are conditional probabilities of values of $X \in V_H$ given those of $\phi^{-1}(X)$, for each $X \in V_H$. One may think of these morphisms as transforming the states of "micro" variables in $V_G$ to those of the corresponding "macro" variables in $V_H$. That these morphisms are consistent with respect to the two functors means that the following diagram commutes for all morphisms (i.e., boxes) $f : X \to Y$ in $\mathsf{Syn}_G$:

$$
\begin{array}{ccc}
F_G(X) & \xrightarrow{F_G(f)} & F_G(Y) \\
\alpha_X \downarrow & & \downarrow \alpha_Y \\
F_H \Phi(X) & \xrightarrow{F_H \Phi(f)} & F_H \Phi(Y)
\end{array}
$$

where the upper half represents a stochastic transition along the causal arrow $f : X \to Y$ in the original graph $V_G$, while the bottom represents the corresponding transition in the coarse-grained graph $V_H$, whereas $\alpha_X$ and $\alpha_Y$ respectively transform the marginal distributions on $F_G(X), F_G(Y)$

of micro variables in $G$ to the marginal distributions on $F_H\Phi(X), F_H\Phi(Y)$ of their macro counterparts. The commutativity of the diagram roughly means that one obtains the same result regardless of whether one follows the causal path in the original model and then transforms the effect (the clockwise path), or transforms the cause first and then calculates its causal consequence in the coarse-grained model (the counter-clockwise path). See Fig. 3 for a numerical illustration.

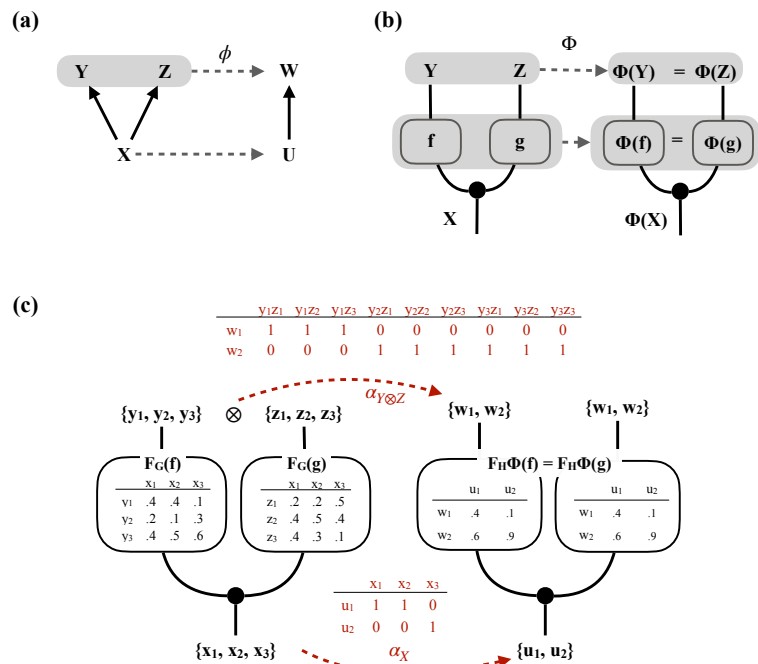

Figure 3: An illustration of $\Phi$-abstraction in three steps. (a) Two DAGs related by graph homomorphism $\phi$, which merges two variables $Y$ and $Z$ into the single $W$. (b) String diagram representations of the DAGs, related via functor $\Phi$. While the functor preserves the fork-like structure, the two arms in the right diagram are identical. (c) Causal models related by $\Phi$-abstraction. Models/functors $F_G$ and $F_H$ assign values to the strings and stochastic matrices to the boxes. Each matrix gives conditional probabilities of an effect given its cause. The red dashed arrows denote a natural transformation that collapses the values $\{x_1, x_2\}$ to $\{u_1\}$ and $\{(y_1, z_i)\}(i = 1, 2, 3)$ to $w_1$, represented by the corresponding deterministic matrices (see Sec. 6). Since $F_G(f) \otimes F_G(g)$ satisfies the homogeneity condition (Def. 7) with respect to $\alpha$, the commutativity holds and $\alpha$ is indeed a natural transformation from $F_G$ to $F_H \cdot \Phi$.

The equivalence of causal models is then defined using the above notion of abstraction. In a nutshell, equivalence is a special case of $\Phi$-abstraction where all the morphisms of the natural transformation are isomorphisms:

**Definition 2 ($\Phi$-equivalence)** *Causal models $F_G$ and $F_H$ are $\Phi$-equivalent if there is a natural isomorphism between $F_G$ and $F_H \cdot \Phi$.*

## 4. Intervention

If the notion of a $\Phi$-abstraction is intended to capture the sameness of different models, it should relate interventions on one model to those on the other in a consistent way. In particular, we expect that any intervention on a macro model can be realized by (a set of) intervention(s) on a corresponding micro model, in such a way that manipulating the abstracted macro model on the one hand and abstracting the manipulated micro model on the other hand yield the same outcome. To check this, we now consider how interventions affect two causal models related by a $\Phi$-abstraction.

Following Jacobs et al. (2019), we first define an intervention as a surgery of a string diagram. An intervention on a variable $X \in V_G$ is denoted by $\mathsf{cut}_X$, which removes the box as well as all the incoming wires of $X$ and replaces them with the "intervened state" $\hat{x}$ with no input:

$$\mathsf{cut}_X \left( \begin{array}{c} {\scriptstyle |X} \\ \boxed{x} \\ {\scriptstyle Y_1 | \ldots | Y_k} \end{array} \right) = \begin{array}{c} {\scriptstyle |X} \\ \widehat{\hat{x}} \end{array}$$

and leaves the others boxes and strings intact. The $\mathsf{cut}$ operation thus defined yields an endofunctor $\mathsf{cut}_X : \mathsf{Syn}_G \to \mathsf{Syn}_G$. The marginal distribution of $X$ after intervention is given by $F_G(\hat{x})$ (we thus assume that the model $F_G$ already contains the information about how each variable could be manipulated. At this point we depart from the original formulation of Jacobs et al. (2019), in which possible post-intervention distributions are restricted to the uniform distribution). Then the whole causal model and joint distribution after the intervention are given by composition of $\mathsf{cut}_X$ with the causal model functor: $F_G \cdot \mathsf{cut}_X : \mathsf{Syn}_G \to \mathsf{Stoch}$.

Next, we consider relating interventions on different models, by embedding interventions on the high-level model $H$ to those on the low-level model $G$. For this purpose, note that the set of all $\mathsf{cut}$ operations on a given diagram, say $\mathsf{Syn}_G$, forms a commutative monoid with the null intervention $\mathsf{cut}_\emptyset$ (which does not change anything) as the identity element and the following composition:

$$\mathsf{cut}_Y \cdot \mathsf{cut}_X = \mathsf{cut}_{\{Y,X\}}$$

that is, intervening on $X$ and then $Y$ amounts to intervening on $\{X, Y\}$ simultaneously. We denote this monoid of interventions on $\mathsf{Syn}_G$ by $\mathsf{cut}(G)$.

With a graph homomorphism $\phi : G \to H$, a macro intervention on $\boldsymbol{X'} \subset V_H$ can be expressed as a combination of micro interventions via the following function:

$$\phi^* :: \mathsf{cut}_{\boldsymbol{X'}} \mapsto \mathsf{cut}_{\phi^{-1}(\boldsymbol{X'})}$$

where $\phi^{-1}(\boldsymbol{X'})$ is the (possibly empty) inverse image of $\boldsymbol{X'}$ under $\phi$, i.e. $\{X \in V_G | \phi(X) \in \boldsymbol{X'}\}$. It is easy to see that $\phi^* : \mathsf{cut}(H) \to \mathsf{cut}(G)$ is a monoid homomorphism, such that

$$\phi^*(\mathsf{cut}_{\boldsymbol{Y'}} \cdot \mathsf{cut}_{\boldsymbol{X'}}) = \phi^*(\mathsf{cut}_{\boldsymbol{Y'}}) \cdot \phi^*(\mathsf{cut}_{\boldsymbol{X'}})$$

for all $\boldsymbol{X'}, \boldsymbol{Y'} \subset V_H$. This leads to the following lemma:

**Lemma 3** *For any* $\boldsymbol{X'} \subset V_H$,
$$cut_{\boldsymbol{X'}} \cdot \Phi = \Phi \cdot \phi^*(cut_{\boldsymbol{X'}}).$$

That is, applying an intervention surgery to the abstracted diagram (LHS) and abstracting the modified diagram (RHS) yield the same string diagram. The commutativity confirms that the modifications $\phi^*(\mathsf{cut}_{\boldsymbol{X'}})$ of $\mathsf{Syn}_G$ and $\mathsf{cut}_{\boldsymbol{X'}}$ of $\mathsf{Syn}_H$ are consistently related via the transformation functor $\Phi : \mathsf{Syn}_G \to \mathsf{Syn}_H$.

With this, we can show that interventions on two models related by a $\Phi$-abstraction yield consistent outcomes.

**Theorem 4** *If $F_H$ is a $\Phi$-abstraction of $F_G$, there is a natural transformation*

$$F_G \cdot \phi^*(\mathsf{cut}_{\boldsymbol{X'}}) \Rightarrow F_H \cdot \mathsf{cut}_{\boldsymbol{X'}} \cdot \Phi$$

*for any $\mathsf{cut}_{\boldsymbol{X'}} \in \mathsf{cut}(H)$.*

**Proof** From Lemma 3, $F_H \cdot \mathsf{cut}_{\boldsymbol{X'}} \cdot \Phi = F_H \cdot \Phi \cdot \phi^*(\mathsf{cut}_{\boldsymbol{X'}})$. Then the natural transformation $\alpha : F_G \Rightarrow F_H \cdot \Phi$ gives the desired natural transformation. In particular, the post-intervention distribution $F_H(\hat{x}')$ for each intervened macro variable $X' \in \boldsymbol{X'}$ is given via the push-forward measure $\alpha_{\boldsymbol{X}} F_G(\hat{\boldsymbol{x}})$, where $\boldsymbol{X} := \phi^{-1}(X')$ is the set of micro variables that constitute $X'$. ∎

Theorem 4 claims that if two models are the "same" in the sense that one is a $\Phi$-abstraction of the other, interventions on the macro model can be represented as those on the micro model, and they yield consistent outcomes. In other words, regardless of whether one intervenes on the micro model and transforms the outcome, or one transforms variables first and then intervenes on the macro model, the result will be the same.

## 5. Comparison with Existing Approaches

$\Phi$-abstraction requires the consistency of each cause-effect connection between two causal models. This is in contrast to existing approaches, where the transformation of causal models is defined with respect to a particular set of interventions. Rubenstein et al. (2017), for instance, define their *exact $\tau$-transformation* as the commutativity of the joint probability distribution along a partial order of interventions. For SEM models $M_G, M_H$ with variable sets $V_G, V_H$, respectively, $M_H$ is said to be an exact $\tau$-transformation of $M_G$ with a variable mapping $\tau : V_G \to V_H$, if there are partially ordered sets (posets) of interventions $\mathcal{I}_G, \mathcal{I}_H$ on $M_G, M_H$ respectively, and a surjective order-preserving map $\omega : \mathcal{I}_G \to \mathcal{I}_H$ such that

$$P^{do(i)}_{\tau(V_G)} = P^{do(\omega(i))}_{V_H}, \ \ \forall i \in \mathcal{I}_G$$

that is, the distribution that results from applying the intervention $i$ on $M_G$ and then the transformation $\tau$ (LHS) is the same as the one obtained by applying the variable transformation and then the intervention $\omega(i)$ on $M_H$ (RHS).

It can be shown, within the limit of finite non-parametric models, that our $\Phi$-abstraction implies an exact $\tau$-transformation:

**Corollary 5** *Let $F_H$ be a $\Phi$-abstraction of $F_G$, and $M_H$ and $M_G$ the corresponding Bayesian networks. Then $M_H$ is an exact $\tau$-transformation of $M_G$.*

*Sketch of proof.* Since the image of $\phi^*$ forms a submonoid in $\mathsf{cut}(G)$, its monoid action yields a partial order of interventions on $G$ (because the monoid operation is defined by union of subsets). Let $\mathcal{I}_G$ be one of such posets. Then $\omega : \mathsf{cut}_X \to \mathsf{cut}_{\phi(X)}$ and $\mathcal{I}_H := \{\omega(\mathsf{cut}_X)|\mathsf{cut}_X \in \mathcal{I}_G\}$ give a bijective mapping $\mathcal{I}_G \to \mathcal{I}_H$. The variable map $\tau$ is given by a natural transformation $\alpha$ such that $\tau : P(X) \mapsto \alpha_X \cdot P(X)$ for any marginal distribution $P(X)$ on a variable $X$ in $G$. Then Theorem 4 guarantees the commutativity of the joint probability distribution.

Conversely, an exact $\tau$-transformation does not imply a $\Phi$-abstraction. Beckers and Halpern (2019, Example 3.4) has shown that Rubenstein et al.'s criterion counts models with different causal graphs as being related by an exact $\tau$-transformation under a restricted range of allowed interventions or probability distributions. Our $\Phi$-abstraction is not liable to such counterexamples, for it requires that models have homomorphic causal graphs.

Beckers and Halpern (2019) and Beckers et al. (2020) take a similar approach, but they restrict the macro-level interventions $\mathcal{I}_H$ to those induced from the micro-level ones $\mathcal{I}_G$ via the variable transformation $\tau$. With this restriction, a pair $(M_H, \mathcal{I}_H)$ of macro-level model-interventions is said to be a $\tau$-*abstraction* of a micro-level pair $(M_G, \mathcal{I}_G)$ if the intervention and transformation commute.

Our $\Phi$-abstraction *partially* satisfies the conditions of a $\tau$-abstraction. Given a set $\mathcal{I}_G$ of micro-level interventions, we can construct a set $\mathcal{I}_H$ of macro-level interventions as in the proof sketch of corollary 5 above. Then, the same corollary guarantees the desired commutativity. Precisely speaking, however, this does not yet give a $\tau$-abstraction, because while Beckers and Halpern (2019) requires the mapping $\tau$ to be surjective, there is no corresponding restriction on the natural transformation $\alpha$ in our framework.

A salient feature of our notion of a $\Phi$-abstraction compared to previous approaches is that it is defined with respect to causal models, independently of any particular sequence or set of interventions. Since a causal model contains the entire intervention calculus within it (in terms of monoid actions as discussed in the previous section), a global correspondence over entire models guarantees a match along any particular sequence of interventions, as long as they are consistently defined.

Another problem with previous approaches is the lack of operationality. That is, given two models, one cannot easily determine whether one is an exact $\tau$-transformation (or $\tau$-abstraction) of the other. Moreover, the previous definitions do not tell us when a given low-level model accommodates a corresponding high-level model. In contrast, in our framework there is a systematic criterion for the existence of a $\Phi$-abstraction, as we discuss below.

## 6. Existence Conditions for a $\Phi$-abstraction

Since a $\Phi$-abstraction is a natural transformation of functors to $\mathsf{Stoch}$, it consists of matrices (i.e., morphisims in $\mathsf{Stoch}$). Hence, to check whether one causal model is a $\Phi$-abstraction of another, it suffices to check the equality of the matrix compositions $F_H \Phi(f_i) \cdot \alpha_{X_i} = \alpha_{X_j} \cdot F_G(f_i)$ for each causal link $f_i : X_i \to X_j$, starting from the exogeneous variables.

Likewise, finding an abstraction of a given causal model boils down to the problem of matrix decomposition, i.e., determining whether for each causal relationship $f : X \to Y$ in the original model, there are transformations $\alpha_X : F_G(X) \to F_H \Phi(X)$ and $\alpha_Y : F_G(Y) \to F_H \Phi(Y)$ such that $\alpha_Y \cdot F_G(f) = g \cdot \alpha_X$ with some stochastic matrix $g : F_H \Phi(X) \to F_H \Phi(Y)$ between the transformed variables. Hereinafter we assume that the models $F_G, F_H$ and the graph homomorphism $\phi$ (and hence the functor $\Phi$) are given, and abbreviate the micro variable $F_G(X)$ as $X$ and macro variable

$F_H\Phi(X)$ as $X'$. Also, we let $f : X \to Y$ denote the stochastic matrix $F_G : F_G(X) \to F_G(Y)$ when no confusion will arise. With this notation, we now ask under what condition such a decomposition is possible, viz., when a given model has a $\Phi$-abstraction or equivalence.

In the case of a natural equivalence, transformations are isomorphisms in Stoch, which are permutation matrices whose rows and columns have the entry 1 in just one place and 0 in all the others.

**Theorem 6** *Causal models $F_G$ and $F_H$ are $\Phi$-equivalent if and only if the translation $\alpha_X : F_G(X) \to F_H \cdot \Phi(X)$ is a permutation for all $X \in V_G$.*

**Proof** Stochastic matrices are invertible if and only if they are permutations. ∎

This means that two models are the same (equivalent) if and only if the variables in one model are a relabeling of those in the other.

In the case of non-equivalent transformations, including abstractions of a low-level to a high-level model, the existence of a matrix decomposition is not guaranteed, except in the following trivial cases:

- The high-level model is a trivial model consisting of singleton variables $\{*\}$ and the trivial identity matrix (scalar) $1 : \{*\} \to \{*\}$.

- $|X'| = |Y|$, with $\alpha_X := F_G(f)$ and $F_H\Phi(f) := \alpha_Y$ for an arbitrary transformation $\alpha_Y : Y \to Y'$. This amounts to interpreting the causal relationships $F_G(f), F_H\Phi(f)$ at each level as if they are "abstractions" of $X$ and $Y$, respectively.

Apart from these trivial cases, the possibility of abstraction generally depends on the nature of the original model, as well as the proposed abstraction.

However, there is a general condition for the existence of a $\Phi$-abstraction when transformations are *deterministic*. Consider a function $\tau : X \to X'$ that maps elements (i.e., values) $\{x_1, \cdots, x_n\}$ of set $X$ to elements $\{x'_1, \cdots, x'_m\}$ of $X'$. Such a function gives rise to a stochastic matrix $\alpha_X$ which has 1 in the $ij$ entry if $\tau(x_j) = x'_i$, and zero otherwise. We call such matrices that are induced by set functions *deterministic transformations*. We also call the inverse images $\tau^{-1}(x'_i) := \{x | \tau(x) = x'_i\}$ the *i-th cell* of $X$ (with respect to $\tau$). Note that a permutation is a deterministic transformation where $\tau$ is bijective. Here we focus on the case where $\tau$ is surjective (and hence $n \geq m$), in which case $X$ has $m = |X'|$ cells that together partition $X$. The induced deterministic transformation $\alpha_X$ then amounts to lumping together the probability masses within each cell of the low-level variable $X$ and equating it with the probability of the corresponding value of the high-level variable $X'$.

Now let us consider, given a low-level causal relationship $f : X \to Y$ and abstracting (i.e., surjective) functions $\tau_X : X \to X'$ and $\tau_Y : Y \to Y'$, whether there is a high-level causal relationship $g : X' \to Y'$ which is a $\Phi$-abstraction. To see this, first note that with appropriate permutations, the deterministic transformations induced by $\tau_X, \tau_Y$ can be diagonalized, and $f$ can be partitioned as follows:

$$\begin{pmatrix} f_1^1 & \cdots & f_m^1 \\ \vdots & \ddots & \vdots \\ f_1^s & \cdots & f_m^s \end{pmatrix}$$

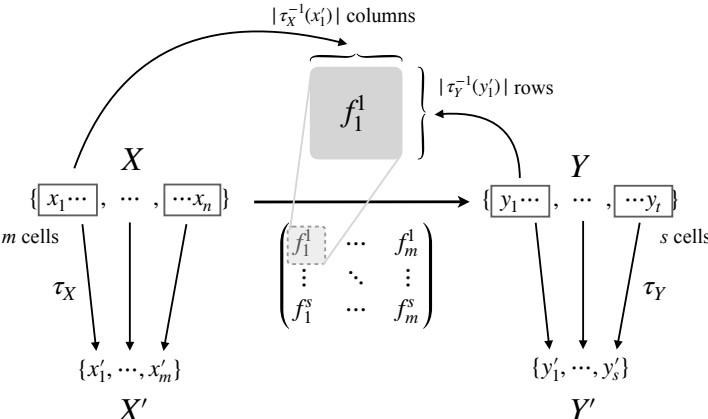

Figure 4: A stochastic matrix $f : X \to Y$ is partitioned into $m \times s$ blocks, where $m$ and $s$ are the numbers of cells in $X$ and $Y$, respectively. The size of each block is determined by the corresponding cells.

where $m$ and $s$ are the numbers of cells in $X$ and $Y$ with respect to $\tau_X$ and $\tau_Y$, respectively, and the size of each partition corresponds to the size of the corresponding cells, so $f_i^j$ is a matrix with $|\tau_X^{-1}(x_i')|$ columns and $|\tau_Y^{-1}(y_j')|$ rows (see Fig. 4).

This prepares us to determine the type of causal relationship that allows for deterministic transformations.

**Definition 7 (causal homogeneity)** $f : X \to Y$ *is* causally homogeneous *with respect to abstracting functions* $\tau_X : X \to X'$ *and* $\tau_Y : Y \to Y'$ *when* $1^T \cdot f_i^j = c_i^j \cdot 1^T$ *for some constant* $c_i^j$ *for every block* $f_i^j$ *of* $f$, *where* $1^T$ *is a unit row vector of an appropriate dimension.*

$1^T \cdot f_i^j$ are the sums of each column of $f_i^j$, where each sum represents the total probabilistic contribution of an element in the $i$-th $X$ cell to the $j$-th $Y$ cell. That this becomes a uniform vector $c_i^j \cdot 1^T$ means that each element within an $X$ cell affects $Y$ cells to exactly the same degree, that is, its causal effects are homogeneous *modulo* cells of the effect variable. The next result shows that this causal homogeneity is a necessary and sufficient condition for the existence of a $\Phi$-abstraction.

**Theorem 8** *Given a causal relationship* $f : X \to Y$ *and abstracting functions* $\tau_X : X \to X', \tau_Y : Y \to Y'$, *there is a (higher-level) causal relationship* $g : X' \to Y'$ *such that* $\alpha_Y \cdot f = g \cdot \alpha_X$ *if and only if* $f$ *is causally homogeneous, where* $\alpha_X$ *and* $\alpha_Y$ *are deterministic transformations induced by* $\tau_X$ *and* $\tau_Y$, *respectively.*

**Proof** See appendix. ∎

Fig. 5 illustrates this with the heart disease example of Fig. 1. The upper layer of the figure is a causal model (i.e., a probabilistic interpretation in Stoch) of the graph $G$ in Fig. 1, while the bottom layer is a model of the graph $H$, where every variable is binary. The proposed abstraction collapses the two cholesterol variables into one via function $\tau : LDL \times HDL \to TC$ with $\tau(l_1, h_1) =$

$\tau(l_1, h_2) = \tau(l_2, h_1) = t_1$ and $\tau(l_2, h_2) = t_2$, keeping the other two variables (Diet and Heart Disease) intact. Whether this function yields a $\Phi$-abstraction depends on the causal homogeneity of $g$, for $f$ is trivially causally homogeneous in this case. Specifically, it must be the case that $g_i^{11} = g_i^{12} = g_i^{21}$ for $i = \{1, 2\}$. This should make sense: since $\tau$ identifies three lower-level combinations $\{(l_1, h_1), (l_1, h_2), (l_2, h_1)\}$ with a single higher-level value $t_1$, these combinations must have the same causal effect on each value $\{y_1, y_2\}$ of HD. The above theorem shows that this is not just a necessary but also a sufficient condition for a given causal model to have an abstraction via deterministic transformations.

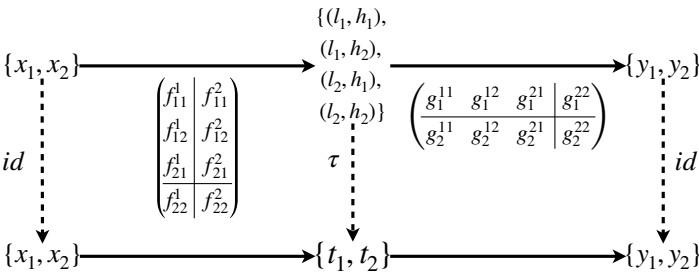

Figure 5: Checking causal homogeneity. The graph schematically shows two causal models (i.e., interpretations in Stoch) based on the graph $G$ (upper) $H$ (bottom) in Fig. 1. The abstracting function $\tau$ partitions stochastic matrices $f$ and $g$ as shown.

## 7. Discussion

This paper has proposed a category-theoretic criterion of equivalence for two causal models with homomorphic DAGs. The basic premise of our approach is that in order for two causal models to be regarded as the same, they must at least capture the same cause-effect relationships, or in other words, their DAGs $G$ and $H$ must be graph-homomorphic. In the string diagram rendition of DAGs, this graph homomorphism $\phi : G \to H$ induces a functor $\Phi : \mathsf{Syn}_G \to \mathsf{Syn}_H$ between the corresponding string diagram categories. Since causal models are identified with functors from a string diagram category to the category Stoch of finite sets and stochastic matrices (Jacobs et al., 2019), the "sameness" of two causal model functors $F_G : \mathsf{Syn}_G \to \mathsf{Stoch}$ and $F_H : \mathsf{Syn}_H \to \mathsf{Stoch}$ can be defined by the natural transformation (or isomorphism) $F_G \Rightarrow F_H \Phi$. If there is such a natural transformation, i.e., when $F_H$ is a $\Phi$-abstraction of $F_G$, the causal flows in the original/low-level model $F_G$ commute with the abstracting transformation, so that they are consistently preserved in the target/high-level model $F_H$. Moreover, interventions on the target model can be translated back into the "constituting" interventions on the original model in such a way that they yield consistent outcomes. Finally, we showed that a given model has a deterministic $\Phi$-abstraction if and only if every causal relationship in the model satisfies the particular condition called causal homogeneity with respect to the proposed abstraction.

Conventional DAGs describe a causal structure as a system of variables connected via arrows, where an arrow $X \to Y$ means that $X$ is a direct cause of $Y$. However, the graph formalism does not specify how the causing takes place. In particular, it does not distinguish whether there are

one, two, or more routes through which $X$ affects $Y$. In this sense, an arrow in a DAG is akin to the notion of *provability* in logic, which just shows that a certain proposition is derivable from another without identifying any particular proof, many of which may exist. In contrast, the categorical approach regards a causal structure as a system of mechanisms (boxes) connected via messengers (strings), or to use a more mundane analogy, factories connected by distribution chains. Here, causation means that some product (string) is transformed into another, and how this transformation is effected is explicitly represented by the mediating boxes/mechanisms (if we stick to the above logical analogy, each box here represents a specific proof). This is the reason why the fork-like structure was preserved in the abstracted string diagram in Fig. 3: although the abstracted model identifies the two arms of the fork, it still retains the information that there are nevertheless two distinct routes. Our observation is that this information, which is lost in a graph-theoretic transformation (homomorphism), is essential for the step-wise comparison between two distinct models, and thus for deciding whether they capture the same causal structure.

Another feature of our approach is that it models the intervention calculus as monoid actions on a causal model. From this perspective, each causal model defines a monoid that encodes a law specifying changes in distribution in response to potential interventions. If two causal models are models of the same physical system, the intervention laws they entail must be consistent. This consistency of the intervention calculus is expressed as a monoid homomorphism, whose existence is guaranteed if the models are related by a $\Phi$-abstraction. The consistency of any particular sequence of interventions is then automatically derived from this global consistency.

From a broader perspective, the category-theoretic approach places causal models in the context of *process theory* and monoidal categories (Coecke and Kissinger, 2017; Jacobs et al., 2019). A further investigation of this connection, as well as an extension of the present approach to continuous variables, should be interesting tasks for future research.

## Acknowledgments

We thank Jimmy Aames for proofreading the manuscript and Tatsuya Yoshi, the editors, and three anonymous reviewers for helpful comments. JO's work was partially supported by JSPS KAKENHI (19K00270) and HS's work was partially supported by JSPS KAKENHI (19K03608 and 20H00001).

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

## Appendix A.  Proof of Theorem 8

Let $|X| = n, |X'| = m, |Y| = t, |Y'| = s$, with $n \geq m$ and $t \geq s$. Suppose that there is a $g : X' \to Y'$ such that $\alpha_Y \cdot f = g \cdot \alpha_X$. By diagonalization and partition we have

$$
\alpha_Y \cdot f = \begin{pmatrix} 1^T & \cdots & 0 \\ \vdots & \ddots & \vdots \\ 0 & \cdots & 1^T \end{pmatrix} \begin{pmatrix} f_1^1 & \cdots & f_m^1 \\ \vdots & \ddots & \vdots \\ f_1^s & \cdots & f_m^s \end{pmatrix} = \begin{pmatrix} 1^T \cdot f_1^1 & \cdots & 1^T \cdot f_m^1 \\ \vdots & \ddots & \vdots \\ 1^T \cdot f_1^s & \cdots & 1^T \cdot f_m^s \end{pmatrix},
$$

and

$$
g \cdot \alpha_X = \begin{pmatrix} g_1^1 & \cdots & g_m^1 \\ \vdots & \ddots & \vdots \\ g_1^s & \cdots & g_m^s \end{pmatrix} \begin{pmatrix} 1^T & \cdots & 0 \\ \vdots & \ddots & \vdots \\ 0 & \cdots & 1^T \end{pmatrix} = \begin{pmatrix} g_1^1 \cdot 1^T & \cdots & g_m^1 \cdot 1^T \\ \vdots & \ddots & \vdots \\ g_1^s \cdot 1^T & \cdots & g_m^s \cdot 1^T \end{pmatrix}.
$$

Hence the identity entails $1^T \cdot f_i^j = g_i^j \cdot 1^T$ for $1 \leq i \leq m$ and $1 \leq j \leq s$. Since $g_i^j$ is a scalar and thus the right hand side is a uniform vector, this means that $f$ is causally homogeneous.

Conversely, if $f$ is causally homogeneous (i.e., $1^T \cdot f_i^j = c_i^j \cdot 1^T$ for a scalar $c_i^j$ for every block $f_i^j$ of $f$), we can set each $g_i^j = c_i^j$. Then by the above matrix calculation we have $\alpha_Y \cdot f = g \cdot \alpha_X$. ∎

