# OpenReview forum: "On the Equivalence of Causal Models: A Category-Theoretic Approach"
_cclear.cc/CLeaR/2022/Conference — CLeaR 2022 Poster_

### Official Review · Reviewer_Lzzj · 2021-11-19

**Confidence:** 3
**Overall Score:** 7

**Main Review:**

Generally, the paper introduces a very elegant method for both thinking and representing abstractions of causal models. While
the mathematical language is somewhat unusual in this field, it is not difficult to understand the main points that the paper
tries to make. I think it also convincingly shows that the approach provides a systematic criterion for the existing of a
causal abstraction. As such, though theoretical, I believe that is a very relevant contribution to this field.

One thing I would like to suggest is the title of the paper: if I understood correctly, the contribution is limited to providing
equivalence between models and their abstractions (the 'micro' and 'macro' level), and not equivalence 'in general' (whatever that
may mean). While discussed in the introduction, I was expecting a somewhat different in the paper at first.


**Summary:**

This paper discusses equivalence between causal models. Somewhat unually, the causal models is represented in a causal string diagram, after which an equivalent abstraction of a causal model is defined by a isomorphism between functors. The motivation for using string diagrams is the allow for preserving different causal mechanisms in the abstraction. The approach is compared to several other approaches for abstractions and it is shown that it generalises those other proposals.

---

> ### Author Response · Authors · 2021-12-03
> **Reply to Reviewer Lzzj**
>
> Thank you for the positive comments. In particular, we are glad to know that the main points of the paper make sense.
>
> We would like to point out that our criterion covers not just micro/macro translations, but all forms of equivalence in general, so that target models may be of completely different physical natures (it could be, for instance, brain networks of humans and macaques). We have to admit that our writing was misleading in this respect. We will emphasize this fact in revision — thank you very much for the pointer.

---

### Official Review · Reviewer_tUp3 · 2021-11-21

**Confidence:** 2
**Overall Score:** 6

**Main Review:**


This paper develops a category-theoretic criterion for determining the equivalence of causal models having different but homomorphic directed acyclic graphs over discrete variables. One of the contributions of this paper is that it interprets a structural causal model in the context of category theory. This allows one to translate a causal diagram into a corresponding string diagram, which appears to be instrumental in determining the homomorphic transformation between different structural causal models.

The authors develop a complete condition for an isomorphic transformation between two causal models using causal string diagrams.  While I have not checked the details of the proof, the results in Theorem 6 seem reasonable. Since I am not familiar with the related work on this specific topic, I cannot comment on the significance and novelty of these results.

Overall, I think this paper studies an interesting problem and proposes a reasonable characterization. However, I do have some questions, summarized as below:
The characterization in Definition 2 and Theorem 7 rely on the detailed parametrization of the underlying causal models. This assumption seems quite strong since these parametrizations are mostly not identifiable in practical applications.

This paper also assumes that all variables in the causal model are observed. Or equivalently, there exists no unobserved confounder (UCs). This brings me back to the previous point: when UCs are present, the underlying causal mechanisms (i.e., the functors) are not generally identifiable from the observational data.

I think the writing of this paper could be improved. I would appreciate one or two possible applications of the proposed method or some practical challenges that involves identifying the homomorphic causal diagrams. This could help the readers to understand the significance of the results.

**Summary:**

Review of Paper#14

---

> ### Author Response · Authors · 2021-12-03
> **Reply to Reviewer tUp3**
>
> Thank you for the comment and question. Yes, we assume that models are parametrized, and have no unobserved confounders. This is because we are interested in the nature of causal models themselves, rather than methods for identifying them from data. Please refer to our reply to reviewer 1 (6giK) for the value of such work.
>
> We admit that the writing could be improved, and agree that some applications might be helpful. But since we have already run out pages, it is difficult to include more examples without sacrificing the rigor and clarity of the theoretical derivations.

---

### Official Review · Reviewer_6giK · 2021-11-22

**Confidence:** 3
**Overall Score:** 7

**Main Review:**

Strengths:
The paper uses category theory to formalize the idea of causal model equivalence which is rather elegant and consistent with previous works in category theory applied to causal modeling.
The theory derivation seems sound.
This model equivalence formalizes the notion of abstraction and thus has applicability to more fundamental ideas like coarse-graining and emergence.

Weaknesses:
The categorical formulation only applies to discrete causal models, whereas in causal ML practice, people more often work with continuous causal models.

The categorical formulation only covers Bayesian networks and not full causal models: In Bayesian networks, only the conditional probabilities of a variable given its parents are known, but not the deterministic function mechanism and the noise specific to this variable, i.e., only P(A|PA_A) is modeled and not the combination of deterministic function and noise A := f(PA_A, N_A).
According to the ladder of causation from Pearl, this makes Bayesian networks only applicable at level 2 of the ladder (interventional), they cannot answer counterfactual (level 3) because counterfactual queries involve knowing the noise and the deterministic function.

It has the strengths and weaknesses of applied categorical approaches: fairly elegant construction that seem to reveal profound structure in the problem of interest, the promise of lifting and porting properties from other areas into the problem of interests via the categorical abstraction. Yet, it appears difficult to find concrete, directly applicable findings. The criteria in section 6. are not easily convertible into operational algorithms.
What does this newly presented derivation allow us to do? What applicable insights does it bring?

I don't think the limitations described above are enough to reject the paper, because building such an understanding of the causal problems is important and might turn out to be useful. In particular, this specific paper on the equivalence of causal models also tackles one particularly important aspect: causal model abstraction which is very related to coarse-graining and emergence.

**Summary:**

The paper proposes a category theory formalization of causal model equivalence, i.e., when one model is an abstraction of another one.

---

> ### Author Response · Authors · 2021-12-03
> **Reply to Reviewer 6giK**
>
> Thank you for the encouraging comment on the value of our work.
>
> Although the present manuscript focuses on discrete cases, we believe that our framework can be extended to continuous cases, given that the category of more general measurable spaces and the Markov kernels between them have categorical structures similar to Stoch. Exploring this possibility will be future work.
>
> We would like to note that our categorical framework CAN separate function and noise, and thus address counterfactual (level 3) inferences. This is easily done by making all box deterministic functions and adding a “noise string” to each box (or more straightforwardly, by constricting a string diagram category from a graph that has explicit noise nodes). We did not mention this in the manuscript because it doesn’t change our theoretical results.
>
> Regarding the lack of operational algorithms, we’d like to stress that this is a proof-of-concept paper, with the view to formulating the concept of the “sameness” of causal structures in mathematically rigorous terms. This is important, because it tells us the identity of causal models — i.e., what we are actually trying to infer with causal discovery algorithms. Prior to any scientific measurement one must know what one is trying to measure. In the case of variables this is specified by identifying their scale, which amounts to specifying admissible transformations---i.e., natural transformations. In this paper we provided a similar criterion for causal models, and thus made it explicit what are those things that are to be discovered by several causal inference algorithms.
>
> Moreover, given our criterion, it is rather straightforward to come up with methods to check whether two models are equivalent. One could check the commutativity (def. 1) by some divergence measure (e.g. KL-divergence) or search coarse-grained models that satisfy causal homogeneity (def. 7). But due to a space constraint, we decided to focus on the conceptual side in this paper.

---

> > ### Comment · Reviewer_6giK · 2021-12-20
> > **Reply to authors**
> >
> > Thank you to the authors for their answer.
> >
> > I think it is correct that the framework can accomodate counterfactuals (level 3). Having many deterministic functions might bring some possible simplifications and new problems with the noises that will be, in practice, unobserved.
> >
> > Overall, I am updating my score a little bit higher as I think the paper deserves to be accepted

---

### Decision · Program_Chairs · 2022-01-12

**Decision:**

Accept (Poster)

**Comment:**

This paper studies some notions of equivalence between and abstraction of causal (Bayesian network) models in category-theoretic terms. The basic idea is to treat a causal model as a functor from a DAG-induced syntax category to the category of stochastic matrices, and define one model being an abstraction of another in terms of the existence of a natural transformation between the functors, and equivalence in terms of the natural transformation being a natural isomorphism. The approach is original, and the writing is reasonably clear for papers of such abstract nature. The main limitation pointed out by reviewers is that it is unclear whether the constructions and results in this paper can yield any application to address more concrete problems. However, all reviewers agree that it is an interesting and novel paper that is suitable for this conference.

In my view, another drawback of the paper is that it omits technical details (due probably to lack of space) that make parts of the paper elusive. For example, when defining the induced syntactical functor on p. 4, the authors seem to suggest that each generating morphism in Syn_{G} is mapped to a generating morphism in Syn_{H}. But it is unclear why this is always feasible. In theory, a graph homomorphism \phi from G to H allows that \phi(Y) has a parent in H that is not \phi(X_k) for any parent X_k of Y in G. When this is the case, it seems that the relevant generating morphism in Syn_{G} would involve more nodes than images of the parents of Y in G. In other words, the image in Syn_{H} of a generating morphism in Syn_{G} is in general not a generating morphism in Syn_{H}, but one constructed from the generators in Syn_{H}. Such a construction seems to be omitted in this paper. I believe it is also relevant when the authors talk about morphisms corresponding to "single causal arrows". I understand that space is limited, but if possible, I suggest the authors should briefly describe the relevant constructions.

Overall I agree with the reviewers that this paper is acceptable.